# Experience-Related Factors in the Success of Beginner Endoscopic Ultrasound-Guided Biliary Drainage: A Multicenter Study

**DOI:** 10.3390/jcm12062393

**Published:** 2023-03-20

**Authors:** Ryota Sagami, Kazuhiro Mizukami, Kazuhisa Okamoto, Chishio Noguchi, Takao Sato, Hidefumi Nishikiori, Yoshinari Kawahara, Masahiro Wada, Yuichiro Otsuka, Satoshi Fukuchi, Hiroshi Takihara, Naosuke Kuraoka, Keita Suzuki, Kazunari Murakami

**Affiliations:** 1Department of Gastroenterology, Oita San-ai Medical Center, 1213 Oaza Ichi, Oita 870-1151, Japan; 2Department of Gastroenterology, Faculty of Medicine, Oita University, 1-1 Idaigaoka, Hasamacho, Yufu 879-5593, Japan; 3Department of Gastroenterology, Nankai Medical Center, 7-8 Tokiwanishimachi, Saiki 876-0857, Japan; 4Department of Gastroenterology, Shin Beppu Hospital, 3898 Tsurumi, Beppu 874-8538, Japan; 5Department of Gastroenterology, Oita Medical Center, 2-11-45 Yokota, Oita 870-0263, Japan; 6Department of Gastroenterology, Oita City Medical Association Almeida Memorial Hospital, 1509-2 Miyazaki, Oita 870-1195, Japan; 7Department of Endoscopy, Kishiwada Tokushukai Hospital, 4-27-1 Kamoricho, Kishiwada, Osaka 596-8522, Japan; 8Department of Gastroenterology, Saiseikai Kawaguchi General Hospital, 5-11-5 Nishikawaguchi, Kawaguchi 332-8558, Japan; 9Department of Computational Brain Imaging, ATR Neural Information Analysis Laboratories, 2-2-2 Hikaridai, Seika-cho, Soraku-gun, Kyoto 619-0288, Japan

**Keywords:** adverse events, choledochoduodenostomy, endoscopic ultrasound-guided biliary drainage, experience, hepaticogastrostomy

## Abstract

Endoscopic ultrasound-guided biliary drainage (EUS-BD) has become comparable to endoscopic retrograde cholangiopancreatography and is now considered a first-line intervention for certain biliary obstructions. Although analysis of experience-related factors may help achieve better outcomes and contribute to its wider adoption, no concrete evidence exists regarding the required operator or institutional experience levels. This study aimed to analyze experience-related factors at beginner multicenters. Patients who underwent EUS-BD using self-expandable metal stents and/or dedicated plastic stents during the study period (up to the first 25 cases since introducing the technique) were retrospectively enrolled from seven beginner institutions and operators. Overall, 90 successful (technical success without early adverse events) and 22 failed (technical failure and/or early adverse events) cases were compared. EUS-BD-related procedures conducted at the time of applicable EUS-BD by each institution/operator were evaluated. The number of institution-conducted EUS-BD procedures (≥7) and operator-conducted EUS screenings (≥436), EUS-guided fine-needle aspirations (FNA) (≥93), and EUS-guided drainages (≥13) significantly influenced improved EUS-BD outcomes (*p* = 0.022, odds ratio [OR], 3.0; *p* = 0.022, OR, 3.0; *p* = 0.022, OR, 3.0; and *p* = 0.028, OR, 2.9, respectively). Our threshold values, which significantly divided successful and failed cases, were assessed using receiver operating characteristic curve analysis and may provide useful approximate indications for successful EUS-BD.

## 1. Introduction

Endoscopic retrograde cholangiopancreatography (ERCP) with transpapillary stent placement is a first-line treatment for both benign and malignant biliary obstructions [1,2]. Recently, the use of endoscopic ultrasound-guided biliary drainage (EUS-BD) has increased as an alternative treatment for patients undergoing difficult ERCP owing to malignant luminal obstruction, surgically altered anatomy, and cannulation failure [3,4]. Moreover, EUS-BD may be superior to percutaneous drainage thanks to its high success rate, resulting in high quality of life and a low risk of adverse events and reinterventions [5,6].

EUS-BD with transluminal stenting, including EUS-guided hepaticogastrostomy (EUS-HGS), choledochoduodenostomy (EUS-CDS), and antegrade biliary stenting (EUS-ABS), is recommended, especially in patients with malignant biliary obstructions; its effectiveness may be comparable with that of ERCP as the first-line management of certain biliary obstructions [7,8,9,10,11]. Since its first description in 2001 [12], the procedure has improved and shown high technical and clinical success rates of approximately 95% and 92–94%, respectively [3,10]. Considering the prevention of adverse events, metal stents [3,4] and newly developed dedicated plastic stents for EUS-BD are being used more widely than conventional plastic stents [13]. In addition, organized technical tips have been designed and widely recognized [4,14].

A recent review reported a relatively lower adverse event rate of 9.3–16% for EUS-BD than that reported in previous studies [3,6,10,15]. As the technique becomes standardized and safer, EUS-BD should be more widely and routinely employed, and its introduction may become possible in beginner centers. As EUS-BD is generally required in only 0.6% of patients with a native papilla undergoing ERCP [16], a more efficient introduction of EUS-BD is required in beginner centers. A thorough analysis of the factors involved in achieving better EUS-BD outcomes and technical success without adverse events may contribute to wider employment of the procedure [17]. Particularly, factors related to experience with the EUS-BD procedure may affect its success. Previous studies reported that technique-related factors may influence the rate of adverse events; however, there is no concrete evidence regarding experience-related factors required by beginner institutions and operators to achieve better EUS-BD outcomes [3,18,19]. Therefore, this multicenter study aimed to clarify potential experience-related factors for achieving better EUS-BD outcomes in beginner multicenters.

## 2. Materials and Methods

### 2.1. Eligibility Criteria

We searched the electronic medical records of seven participating institutions to enroll patients undergoing EUS-BD with transmural stenting as an alternative treatment for ERCP due to malignant luminal obstruction, surgically altered anatomy, ERCP failure, and insufficient first drainage by ERCP between January 2014 and June 2021. The subtypes of the EUS-BD procedures included in this multicenter study were EUS-HGS, HGS combined with ABS, CDS using a self-expandable metal stent (SEMS), and EUS-BD with dedicated plastic stents. EUS-guided rendezvous procedures were excluded. The participating institutions were relatively low-volume centers with ≤500 ERCP cases per year, no previous experience performing EUS-BD, and only one beginner EUS-BD operator in each. The first 25 patients from each institution admitted since the introduction of EUS-BD were enrolled. All operators had previously performed >500 ERCP procedures at the time of their first EUS-BD; additionally, they had studied the EUS-BD procedures by observation in high-volume centers before their first EUS-BD. Most assistants had little to no experience with EUS-BD or assisting with it. All institutions had surgeons or radiologists standing by to handle a potential emergency.

This study was designed and conducted in accordance with the ethical guidelines of the 1975 Declaration of Helsinki (6th revision, 2008) and was registered by the University Hospital Medical Network Clinical Trials Registry (UMIN000047698) as a retrospective study after receiving approval from our Institutional Review Board (IRB protocol number, 2021001K). The need for informed consent was waived owing to the retrospective nature of the study.

### 2.2. Definition and Study Outcome

This study compared two groups of patients undergoing EUS-BD: successful (technical success without early adverse events) and failed (technical failure and/or early adverse events) cases. Technical success was defined as successful stent placement between the biliary tree and the gastrointestinal lumen. Early adverse events were defined as procedure-related adverse events occurring within 2 weeks of stent placement that required endoscopic, surgical, or radiological intervention, or additional conservative treatment. The adverse events included bleeding, cholangitis, bile peritonitis, and pneumoperitoneum. The severity of adverse events was evaluated according to the American Society of Gastrointestinal Endoscopy classification [20]. Figure 1 depicts the decision-making process for the distinction between successful and failed cases.

Overall, 112 patients who underwent EUS-BD were enrolled, and 90 patients who underwent technically successful EUS-BD and completed treatment without early adverse events were classified as successful. The technical success rate was 96.4%, and the early adverse event rate was 14.3%. A total of 22 patients, including 6 patients who experienced technical failure and 16 patients who experienced technical success with early adverse events, were classified as failed cases.

Experience with EUS-BD was defined as the number of procedures conducted by each institution and operator by the time of the applicable EUS-BD procedure. The operators’ EUS-related procedural experiences were evaluated by analyzing the number of pancreatobiliary screening EUS, EUS-guided fine-needle aspiration (FNA), and other EUS-guided drainage procedures (cyst, abscess, and gallbladder) that had been conducted by the time of applicable EUS-BD procedure introduction.

The primary aim of the study was to analyze the experience-related factors of EUS-BD/EUS-BD-related procedures for beginner institutions and operators that may affect the occurrence of successful cases. The secondary aim was to clarify the significant procedure-related factors involved in the occurrence of successful cases.

### 2.3. Statistical Analyses

Patient characteristics, experience-related factors, and procedure-related factors were compared between successful and failed cases. As appropriate, categorical parameters were compared using the chi-square test or Fisher’s exact test. Continuous variables are expressed as mean ± standard deviation, depending on the normality of the distribution. The thresholds for EUS-BD and EUS-BD-related procedures that significantly divided successful and failed cases were determined using the receiver operating characteristic curve (ROC) analysis. The threshold was defined as the cross point of the true and false positive rates of the ROC. The odds ratios (OR) between the thresholds were further evaluated with 95% confidence intervals (CI). All statistical analyses were conducted using SPSS version 28.0 (IBM Corp., Armonk, NY, USA). Statistical significance was set at *p* < 0.05.

## 3. Results

Table 1 summarizes the patients’ characteristics for the successful and failed cases. Other than sex, there are no significant differences between the groups, as shown in Table 1.

Table 2 summarizes the number of EUS-BD and EUS-BD-related procedures conducted by each institution and operator at the time of the applicable EUS-BD procedure.

The threshold of EUS-BD conducted by an institution that significantly divided the successful and failed cases was seven cases. Similarly, the threshold of EUS-BD conducted by an operator was six cases, and that of EUS screening, EUS-FNA, and EUS-guided drainage conducted by the operator were 436, 93, and 13 cases, respectively.

The number of EUS-BD cases conducted by the institution (≥7) is a significant factor in the occurrence of successful cases (63.6% vs. 36.7%; *p* = 0.022; OR, 0.33 [95% CI, 0.13–0.87]). Conversely, the number of EUS-BD cases conducted by the operator (≥6) is not a significant factor for the occurrence of successful cases (54.5% vs. 40.0%; *p* = 0.217; OR, 0.56 [0.22–1.42]). In addition, the number of EUS screenings cases (≥436), EUS-FNA cases (≥93), and EUS-guided drainage cases (≥13) conducted by the operator are all significant factors for successful cases (63.6% vs. 36.7%, *p* = 0.022, OR, 3.0 [1.1–8.0]; 63.6% vs. 36.7%, *p* = 0.022, OR, 3.0 [1.1–8.0]; and 63.6% vs. 37.8%, *p* = 0.028, OR, 2.9 [1.1–7.6], respectively). Only 13 EUS-BD procedures were conducted by assistants who had already performed or assisted with ≥6 EUS-BD cases; however, the experience is not significant for the occurrence of successful cases (13.3% vs. 4.5%, *p* = 0.249).

Table 3 summarizes the procedure-related factors. Naturally, the location of the biliary obstruction/puncture site for EUS-CDS was the extrahepatic bile duct/common bile duct. Among the subtypes of EUS-BD, location of biliary obstruction, assistant (non-doctor or doctor), puncture site, puncture bile duct diameter, type of guidewire, type of fistula dilation, stent diameter/length, and location of biliary obstruction (extrahepatic bile duct obstruction) were the only significant factors involved in successful cases (73.3% vs. 40.9%; *p =* 0.004; OR, 4.0 [1.5–10.5]).

Table 4 summarizes the details of technical failures and early adverse events of failed cases.

## 4. Discussion

The results of this study performed at EUS-BD beginner institutions with relatively similar conditions suggest that various experience-related factors (number of EUS-BD conducted by the institution and EUS screening, EUS-FNA, and EUS-guided drainage conducted by the operator) may affect the occurrence of successful EUS-BD outcomes, whereas the operator’s experience with EUS-BD itself may not.

Some studies analyzed the learning curve of EUS-BD of a single operator in a high-volume center [21,22]. They recommend that 7–32 cases are required to achieve stable, successful EUS-BD or a reduction in procedural time [21,22,23]. Another study of 101 patients reported that the first half of enrolled patients had a lower mortality rate than the second half [24]. In another study, the success rate in the first 3 years was lower than that in the last 2 years [25]. The analysis of the learning curve or the period of gaining EUS-BD experience was reported for some institutions; however, most previous studies were conducted in high-volume centers and performed by a single expert operator. In addition, a learning curve was defined to achieve quicker EUS-BD [21].

Conversely, other studies reported that the accumulation of operator experience over time did not yield significant reductions in EUS-BD-related adverse events [17,26]. Moreover, experience with EUS-guided interventions may be associated with fewer EUS-BD-related adverse events due to the technical similarities between EUS-guided pseudocyst drainage and EUS-BD [27]. The findings of these reports support our results.

It has been recommended that EUS-BD should be attempted only by experts in high-volume centers with appropriate training [3] owing to its lower technical success rate [17], higher adverse event rate compared with ERCP [3], and seriousness of technical failures/adverse events [24,28]. In contrast, a review showed a relatively lower adverse event rate of 9.3–16% for EUS-BD [10,15] compared with that of 23% reported previously [3]. In addition, increased success rates and decreased adverse events were observed in studies published after 2013, probably because of the newly designed safer stents [29,30], organized technical tips/algorithms for EUS-BD [4,14], and greater experience of endoscopists [10]. Considering these facts, the widespread introduction of EUS-BD may be feasible in appropriate conditions. Certainly, EUS-BD should be performed with surgical and radiological backup [3].

Worldwide societies of gastrointestinal endoscopy recommend that several cases of EUS screening and EUS-FNA should be performed for learning EUS procedures [19,31]. EUS-BD trainees should be trained in convex EUS handling and be allowed to simulate the procedures using simulators and models for EUS-BD before EUS-BD implementation [32]. They should begin the EUS-BD procedures under expert supervision and, if possible, complete training programs in experienced institutions [18]. One expert center recommends that at least 20 EUS-BD procedures should be performed by trainees under expert supervision [32]. However, it is difficult for all institutions to perform EUS-BD under similar conditions. While step-by-step training for EUS-BD and actual procedures under supervision are essential, a more efficient EUS-BD introduction model is required.

Our results suggest that the institution-conducted experience with EUS-BD and operator-conducted EUS-BD-related procedures have a greater influence on better outcomes than the operator-conducted EUS-BD experience, indicating the importance of the institution-conducted experience, likely because EUS-BD experience may reflect the experience of the assistant doctors and engineers in addition to that of the operators. Further, operator experience in EUS-BD-related procedures may compensate for the operator’s lack of EUS-BD experience itself. The threshold values calculated in this study may be used as a rough indicator for the achievement of successful EUS-BD outcomes at a reasonable number of procedures.

Regarding procedure-related factors, one study reported fistula dilation with a needle knife as a significant factor responsible for adverse events [26]. However, no specific procedure-related factors were confirmed in another systematic review [3]. This study showed that the predictor of the bile duct obstruction location (extrahepatic bile duct obstruction compared to intrahepatic obstruction) was only relevant in successful cases. A study of the initial experience with EUS-BD reported that manipulating the guidewire into the duct was the most difficult aspect of the procedure [17]. Manipulating the guidewire across the intrahepatic bile duct may be more difficult than crossing the extrahepatic duct, which may have affected the results of this study. A beginner operator may need to be careful in EUS-BD procedures for intrahepatic bile duct obstruction cases.

Comprehensively, from our results, a certain level of institutional experience with EUS-BD and operator experience with EUS-BD related procedures may be required for the occurrence of a successful case for the very first EUS-BD procedure performed by a beginner. In addition, EUS-BD beginners may need to avoid intrahepatic obstruction cases.

This study has the following strengths. To the best of our knowledge, this is the first multicenter study to report on the various experience-related factors that lead to better EUS-BD outcomes with the corresponding threshold values. In addition, participating beginner institutions had relatively similar conditions. However, this study also has some limitations. The retrospective design, the variation in the number of cases, and the dataset validation between institutions may have introduced selection bias. In addition, the indications for the procedure and technical methods for EUS-BD were not completely unified among the institutions. Ideally, experience- and procedure-related factors should be evaluated for each EUS-BD procedure because of the slight differences between them. In the future, further studies with a greater number of institutions under similar procedural conditions and of enrolled cases should be carried out.

## 5. Conclusions

The number of institution-conducted EUS-BD procedures and operator-conducted EUS screening, FNA, and EUS-guided drainage led to significantly better EUS-BD outcomes at beginner centers. Our threshold values may provide reasonable and approximate indications for successful EUS-BD.

## Figures and Tables

**Figure 1 jcm-12-02393-f001:**
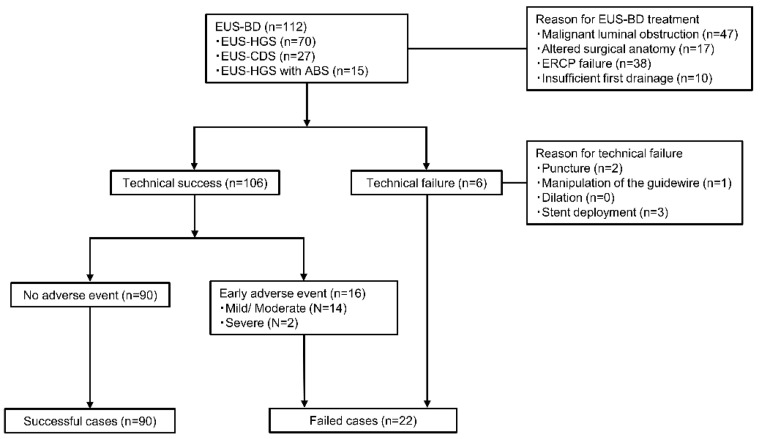
Flow diagram of the study design. ABS, antegrade biliary stenting; BD, biliary drainage; CDS, choledochoduodenostomy; ERCP, endoscopic retrograde cholangiopancreatography; EUS, endoscopic ultrasound; HGS, hepaticogastrostomy.

**Table 1 jcm-12-02393-t001:** Patient characteristics in successful and failed cases.

	Successful Cases (n = 90)	Failed Cases (n = 22)	*p*-Value
Age, mean (±SD), y	76.0 ± 12.8	75.5 ± 13.9	0.822
Male sex, n (%)	42 (46.7)	17 (77.2)	0.010
Etiology of biliary obstruction, n (%)			
Malignant	84 (93.3)	18 (81.8)	0.090
Benign	6 (6.7)	4 (18.2)	
Presence of jaundice, n (%)	72 (80.0)	15 (68.2)	0.233
Serum bilirubin (mean ± SD), mg/dL	7.5 ± 6.7	6.5 ± 7.3	0.520
Presence of cholangitis, n (%)	35 (38.9)	9 (40.9)	0.862
C-reactive protein (mean ± SD), mg/dL	5.4 ± 5.4	3.5 ± 4.7	0.249
Presence of ascites, n (%)	15 (16.7)	3 (13.6)	0.729
Reasons for EUS-BD, n (%)			0.301
Malignant luminal obstruction	38 (42.2)	9 (40.9)	
Altered surgical anatomy	11 (12.2)	6 (27.3)	
Failed ERCP	33 (36.7)	5 (22.7)	
Insufficient previous drainage	8 (8.9)	2 (9.1)	

EUS-BD, endoscopic ultrasound-guided biliary drainage; ERCP, endoscopic retrograde cholangiopancreatography; SD; standard deviation.

**Table 2 jcm-12-02393-t002:** Number of EUS procedures conducted and institution and operator experience-related factors.

	Successful Cases (n = 90)	Failed Cases (n = 22)	*p*-Value	Odds Ratio (95% CI)
Conducted EUS-BD (mean ± SD)			NA	NA
Institution	10.1 ± 6.7	6.1 ± 4.7		
Operator	9.0 ± 7.1	5.6 ± 5.0		
Conducted EUS-BD, n (%)				
Institution ≥ 7 cases	57 (63.3)	8 (36.4)	0.022	3.0 (1.1–8.0)
Operator ≥ 6 cases	54 (60.0)	10 (45.5)	0.217	1.8 (0.7–4.6)
Operator-conducted EUS-BD related procedures (mean ± SD)			NA	NA
EUS screening	712.1 ± 422.1	586.4 ± 487.4		
EUS-FNA	133.7 ± 96.9	82.6 ± 54.9		
EUS-guided drainage	17.1 ± 10.3	10.9 ± 8.5		
Operator-conducted procedures, n (%)				
Screening EUS ≥ 436 cases	57 (63.3)	8 (36.4)	0.022	3.0 (1.1–8.0)
EUS-FNA ≥ 93 cases	57 (63.3)	8 (36.4)	0.022	3.0 (1.1–8.0)
EUS-guided drainage ≥ 13 cases	56 (60.0)	8 (36.4)	0.028	2.9 (1.1–7.6)

CI, confidence interval; EUS, endoscopic ultrasound; EUS-BD, endoscopic ultrasound-guided biliary drainage; EUS-FNA, endoscopic ultrasound-guided fine-needle aspiration; NA, not assessed; SD, standard deviation.

**Table 3 jcm-12-02393-t003:** Procedure-related factors.

	Successful Cases (n = 90)	Failed Cases (n = 22)	*p*-Value
EUS-BD, n (%)			0.539
EUS-HGS	54 (60.0)	16 (72.8)	
EUS-CDS	23 (25.6)	4 (18.2)	
EUS-HGS with ABS	13 (14.4)	2 (9.0)	
Location of biliary obstruction, n (%)			0.004 *
Extrahepatic bile duct	66 (73.3)	9 (40.9)	
Intrahepatic bile duct	24 (26.7)	13 (59.1)	
Assistant, n (%)			0.109
Non-doctor	15 (16.7)	7 (31.8)	
Doctor	75 (83.3)	15 (68.2)	
Puncture site, n (%)			0.561
B2	17 (18.9)	3 (13.6)	
B3	50 (55.6)	15 (68.2)	
Common bile duct	23 (25.5)	4 (18.2)	
Puncture bile duct diameter, mean (± SD), mm	8.0 ± 4.5	7.1 ± 3.4	0.239
Guidewire, n (%)			0.198
0.025 inch	76 (84.4)	16 (72.7)	
0.035 inch	14 (15.6)	6 (27.3)	
Fistula dilation, n (%)			0.555
Without dilation	9 (10.0)	4 (18.2)	
Blunt dilation (bougie or balloon)	60 (66.7)	13 (59.1)	
Electrical dilation	21 (23.3)	5 (22.7)	
Type of stent, n (%)			0.756
Dedicated plastic stent	20 (22.2)	3/16 (18.8)	
SEMS	70 (77.8)	13/16 (81.2)	
Stent diameter, n (%)			0.803
7 Fr	20 (22.2)	3/16 (18.8)	
6/8 mm	33 (36.7)	5/16 (31.2)	
10 mm	37 (41.1)	8/16 (50.0)	
Stent length, n (%)			0.158
60/80 mm	30 (33.3)	2/16 (12.5)	
100/120 mm	40 (44.4)	11/16 (68.8)	
140 mm	20 (22.3)	3/16 (18.7)	

* *p* < 0.05, compared to the success and failure cases. EUS-BD, endoscopic ultrasound-guided biliary drainage; EUS-CDS, endoscopic ultrasound-guided choledochoduodenostomy; EUS-HGS, endoscopic ultrasound-guided hepaticogastrostomy; ABS, antegrade biliary stenting; CDS, choledochoduodenostomy; SD, standard deviation; SEMS, self-expandable metal stent.

**Table 4 jcm-12-02393-t004:** Details of technical failure and early adverse events.

Technical failure step	6 (cases)
Puncture	2
Manipulation of the guidewire	1
Dilation	0
Stent deployment	3
Early adverse events	16 (cases)
Peritonitis	9
Bleeding	2
Pneumoperitoneum	2
Cholangitis	2
Pancreatitis	1
Severity of adverse events	
Mild	10
Moderate	4
Severe	2

## Data Availability

The dataset used during the current study is available from the corresponding author upon reasonable request.

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
