# Peer review of "Experience-Related Factors in the Success of Beginner Endoscopic Ultrasound-Guided Biliary Drainage: A Multicenter Study"

_jcm, 2023, doi:10.3390/jcm12062393_

Round 1

Reviewer 1 Report

This useful article focuses on EUS-BD, which is rapidly becoming popular in recent years, in line with actual clinical practice.
Until now, the overwhelming majority of reports have been from high-volume centers, and it is possible that these results were obtained in a well-developed environment. We believe it is very important to consider the skills required for beginner institutions and surgeons to standardize EUS-BD.

Eligibility Criteria

 We believe it is important to define the assistants. Did the assistants in this study have experience with a few cases of EUS-BD or with assisting with EUS-BD, although few? If possible, please provide the criteria for assistants.

 In all cases, if one operator has difficulty completing the procedure, is it considered a failure without being replaced by another operator?

Result.

 In Table 3, "Location of biliary obstruction" is listed as a significant factor, but the site of obstruction is often different between HGS and CDS in the first place. If possible, please list it separately, as in "Puncture site" in Table 3.

Is the pancreatitis a case of ABS? If so, it is possible that there would have been no pancreatitis and the case would have been a success if there had been no ABS.

Finally, if EUS-BD is to be performed for the first time in the future, what kind of cases do you think should be performed first based on the results of this study?

Is HGS better, CDS better, if the site of obstruction is intrahepatic, should it be stopped as the first case, should the assistant be a doctor, etc.?

Author Response

March 16, 2023

Prof. Dr. Maciej Banach

Section Editor-in-Chief

Journal of Clinical Medicine

Manuscript ID: jcm-2240477, R1

Title: Experience-related factors in the success of beginner endoscopic ultrasound-guided biliary drainage: A multicenter study

Dear Editor:

Thank you very much for reviewing our manuscript; we appreciate the reviewers’ constructive comments. We have incorporated the related suggestions that have greatly improved the quality of our manuscript. We hope that the manuscript is now suitable for publication in the Journal of Clinical Medicine. Our point-by-point responses to the reviewers’ comments are indicated below. The related changes in the manuscript are indicated in red text.

Thank you for your consideration. We look forward to hearing from you.

Sincerely,

Kazuhiro Mizukami

Department of Gastroenterology, Faculty of Medicine, Oita University, 1-1 Idaigaoka, Hasamacho, Yufu, Oita 879-5593, Japan

Phone: +81-97-586-6193

Fax: +81-97-586-6194

Comments by reviewer #1 

This useful article focuses on EUS-BD, which is rapidly becoming popular in recent years, in line with actual clinical practice.

Until now, the overwhelming majority of reports have been from high-volume centers, and it is possible that these results were obtained in a well-developed environment. We believe it is very important to consider the skills required for beginner institutions and surgeons to standardize EUS-BD.

Eligibility Criteria

 We believe it is important to define the assistants. Did the assistants in this study have experience with a few cases of EUS-BD or with assisting with EUS-BD, although few? If possible, please provide the criteria for assistants.

Response

Thank you for your constructive comments. Most assistants had little to no experience with EUS-BD or assisting with it, and only a few of them had assisted or performed ≥6 EUS-BD procedures. However, assisting or performing ≥6 EUS-BD procedures was not a significant factor in the occurrence of successful cases. To address your comment, we have added relevant sentences to our Materials and Methods and Results sections as shown below.

Page 2, lines 48 to 49

“Most assistants had little to no experience with EUS-BD or assisting with it.”

Page 5, lines 137 to 139

“Only 13 EUS-BD procedures were conducted by assistants who had already performed or assisted with ≥6 EUS-BD cases; however, the experience was not significant for the occurrence of successful cases (13.3% vs. 4.5%, P=0.249).”

 In all cases, if one operator has difficulty completing the procedure, is it considered a failure without being replaced by another operator?

Result.

 In Table 3, "Location of biliary obstruction" is listed as a significant factor, but the site of obstruction is often different between HGS and CDS in the first place. If possible, please list it separately, as in "Puncture site" in Table 3.

Response

Thank you for your constructive comments. As you mentioned, the location of the biliary obstruction of EUS-CDS was the extrahepatic bile duct. In addition, the EUS-CDS puncture site was the common bile duct. Moreover, the kind of EUS-BD procedure and the puncture site did not significantly contribute to the occurrence of successful cases; conversely, the location of the biliary obstruction was a significant factor in this sense. To address your comment, we have added relevant sentences to our Results and Discussion sections as mentioned below.

Page 5, lines 140 to 142

“Naturally, the location of the biliary obstruction/puncture site for EUS-CDS was the extrahepatic bile duct/common bile duct.”

Page 8, lines 233 to 234

“Ideally, experience- and procedure-related factors should be evaluated for each EUS-BD procedure because of the slight differences between them.”

Is the pancreatitis a case of ABS? If so, it is possible that there would have been no pancreatitis and the case would have been a success if there had been no ABS.

Response

Thank you for your important comments. The case with post-procedure pancreatitis underwent EUS-ABS. As you mentioned, the case might have been classified as a successful case if there had been no ABS. On the other hand, if there had been ABS, the case might have been classified as a failed case because of other adverse events such as a bile leak. Thus, we need to think about a relevant EUS-ABS adaptation.

Finally, if EUS-BD is to be performed for the first time in the future, what kind of cases do you think should be performed first based on the results of this study?

Is HGS better, CDS better, if the site of obstruction is intrahepatic, should it be stopped as the first case, should the assistant be a doctor, etc.?

Response

Thank you for your comprehensive comments. To address these questions, we have added some relevant sentences in the last two paragraphs of our Discussion section as follows.

Page 8, lines 222 to 225

“Comprehensively, from our results, a certain level of institutional experience with EUS-BD and operator experience with EUS-BD related procedures may be required for the occurrence of a successful case for the very first EUS-BD procedure performed by a beginner. In addition, EUS-BD beginners may need to avoid intrahepatic obstruction cases.”

Reviewer 2 Report

The authors present the experience-related factors in the success of EUS-BD. Due to the rising incidence of advanced pancreatic lesions the topic is very interesting and important. 

Overall, the study is well written. However some issues can be found:

line 43: "The participating institutions were excluded" - excluded from what?

line 47: "... > 500 ERCP procedures at the time of their first Z EUS-BD..." - what does it mean - "Z EUS BD"

Author Response

March 16, 2023

Prof. Dr. Maciej Banach

Section Editor-in-Chief

Journal of Clinical Medicine

Manuscript ID: jcm-2240477, R1

Title: Experience-related factors in the success of beginner endoscopic ultrasound-guided biliary drainage: A multicenter study

Dear Editor:

Thank you very much for reviewing our manuscript; we appreciate the reviewers’ constructive comments. We have incorporated the related suggestions that have greatly improved the quality of our manuscript. We hope that the manuscript is now suitable for publication in the Journal of Clinical Medicine. Our point-by-point responses to the reviewers’ comments are indicated below. The related changes in the manuscript are indicated in red text.

Thank you for your consideration. We look forward to hearing from you.

Sincerely,

Kazuhiro Mizukami

Department of Gastroenterology, Faculty of Medicine, Oita University, 1-1 Idaigaoka, Hasamacho, Yufu, Oita 879-5593, Japan

Phone: +81-97-586-6193

Fax: +81-97-586-6194

Comments by reviewer #2

The authors present the experience-related factors in the success of EUS-BD. Due to the rising incidence of advanced pancreatic lesions the topic is very interesting and important.

Overall, the study is well written. However some issues can be found:

line 43: "The participating institutions were excluded" - excluded from what?

Response

Thank you for your detailed feedback. This sentence had no meaning and was a mistake. As such, we have now deleted it. We apologize for this mistake.

line 47: "... > 500 ERCP procedures at the time of their first Z EUS-BD..." - what does it mean - "Z EUS BD"

Response

Thank you for your detailed feedback. The letter Z was a typo, which we have now deleted. We apologize for this mistake.

Lastly, the overall English of the manuscript was re-checked by a native English speaker before the re-submitting to your journal. Please confirm the article.